# Concept Unlearning for Large Language Models

**Tomoya Yamashita    Takayuki Miura    Yuuki Yamanaka**
**Toshiki Shibahara    Masanori Yamada**
NTT Social Informatics Laboratories
{tomoya.yamashita,tkyk.miura,yuuki.yamanaka}@ntt.com
{toshiki.shibahara,masanori.yamada}@ntt.com

## Abstract

Existing studies have reported that corpora on the Web for training large language
models (LLMs) may contain undesirable information such as Personally Identi-
fiable Information, leading to privacy violations when operating LLMs. To deal
with this problem, Machine Unlearning (MU) has attracted attention, aiming to
forget arbitrary information from AI models. However, the existing MU responds
to a deletion request for specific data points in the AI model, and it is difficult to
respond to a deletion request for a specific concept in the LLM (e.g., a person's
name). This paper proposes a new MU requirement called Concept Unlearning
(CU) to make LLMs forget arbitrary concepts from the perspective of a knowledge
graph. This will allow us to define forgetting in terms of "knowledge", which
is more intuitive to humans, and allow us to design effective methods for LLM
forgetting. We also propose a method to realize CU by generating appropriate
token sequences using LLMs and applying gradient ascent on the generated token
sequences. The effectiveness of our method is confirmed by the dataset created
from Wikipedia and LLM-as-a-Judge.

## 1   Introduction

The development of computer technology and AI research has led to remarkable performance
improvements in large language models (LLMs). One of the main factors improving LLMs is the use
of large Web corpora as training data. It has been reported that the size of training data is important
for training LLMs, and training LLMs with large corpora is considered an essential process for
acquiring high-performance LLMs [Hoffmann et al., 2022, Muennighoff et al., 2024]. However,
corpora on the Web may contain undesirable information such as Personally Identifiable Information,
and existing studies have reported that LLMs trained using such training data violate privacy [Carlini
et al., 2019, 2021]. In addition, the EU General Data Protection Regulation (GDPR), which regulates
the handling of personal data, requires LLM providers to respond to data deletion requests from data
suppliers promptly. These issues need to be solved so that LLMs can be socially trusted and used
safely and ethically.

One approach to these challenges is machine unlearning (MU), which aims to forget the information
of arbitrary training data from the AI model [Nguyen et al., 2022]. In the existing MU problem
setting, the following two steps are typically performed.

1. Specify the forgetting target data points in the training dataset.

2. Apply the forgetting process to the AI model on the target data points.

Existing MU researches focus on proposing methods that achieve high forgetting performance on
the target data points, i.e., corresponding to Step 2 [Bourtoule et al., 2021, Ginart et al., 2019, Yao
et al., 2023, Neel et al., 2021]. However, a deletion request to LLMs is not necessarily to delete

38th Conference on Neural Information Processing Systems (NeurIPS 2024).

specific data points (sentences) in the training dataset, but to delete specific concepts (e.g., a person's name). In such a case, it is difficult to conduct Step 1, specifying the forgetting target data points corresponding to the target concept in the training dataset. This is because it is difficult to determine if a sentence is the forgetting target sentence, i.e., should the LLM forget about the father of the target concept, friends, or teachers? Many existing MU methods can achieve high forgetting performance on the forgetting target data points, however, it is out of scope to determine whether each data point is to be forgetting target or not. Therefore, many existing MU methods are difficult to respond deletion requests for specific concepts to LLMs.

To solve this problem in LLM forgetting, we propose Concept Unlearning (CU), which is a new requirement for MU. In CU, we use a knowledge graph (KG) to interpret the knowledge held by LLMs. KG is a network of various types of knowledge and has been used in research on interpreting the knowledge of LLMs [Petroni et al., 2019, Luo et al., 2023]. CU requires removing the knowledge about the forgetting target from the KG of LLMs and does not require specifying the forgetting target data points in the training dataset. Therefore, Step 1 can be designed more flexibly for realizing CU.

In this paper, we design a method to realize CU by applying Gradient Ascent (GA) on appropriate token sequences. In the proposed method, we consider generating the forgetting target token sequences automatically from the KG. By generating from the KG, it is possible to prepare the forgetting target token sequences without excess or deficiency, and Step 1 can be executed appropriately for CU. We use an LLM (reference LLM) to obtain the token sequences corresponding to the forgetting target concept from the KG. Then we apply GA on the forgetting target LLM with the obtained token sequences. In the evaluation experiment, we defined the target concept for forgetting a person's name. We created and used a dataset derived from Wikipedia to evaluate whether our method satisfies the CU requirements. In addition, we evaluate our method by LLM-as-a-Judge using GPT-4o. Through evaluation experiments, we confirm that our method is effective for realizing CU. Also, we confirmed that our method does not interfere with LLMs' general knowledge by evaluating them before and after the forgetting process using 8 datasets that questioned their general knowledge.

## 2 Preliminary

### 2.1 Notation

LLM is defined as a function $\boldsymbol{f_\theta} : V^N \to V^M$ that converts an input token sequence $\boldsymbol{x} \in V^N$ into an output token sequence $\boldsymbol{y} \in V^M$. Here, let $V \subset \mathbb{N}$ denote the vocabulary set of LLM. $V$ is assumed to contain an empty token $\epsilon$. Let $N$ and $M$ be the length of the input and output token sequence, and $\boldsymbol{\theta}$ be the model parameters of LLM. The output of the LLM $y_i = (\boldsymbol{f_\theta}(\boldsymbol{x}))_i$ can be defined as follows.

$$y_i \sim \begin{cases} p_{\boldsymbol{\theta}}(y|\boldsymbol{x}) & (i = 1) \\ p_{\boldsymbol{\theta}}(y|\boldsymbol{x}, y_1, \cdots, y_{i-1}) & (i > 1). \end{cases} \tag{1}$$

In Eq. 1, $p_{\boldsymbol{\theta}}$ is the output probability distribution of the LLM and $1 \leq i \leq M$. As Eq. 1 indicates, LLM generates a token sequence autoregressively by generating the next token based on the input token sequence and inputting the generated tokens back into the model.

### 2.2 Related Works

#### 2.2.1 Knowledge Graph

Knowledge Graph (KG) is a graph structure network of various types of knowledge used for knowledge coordination, integration, and advanced analysis. KG is also used in research to interpret the knowledge held by LLMs [Petroni et al., 2019, Luo et al., 2023]. In KG format, the knowledge possessed by LLM is represented in $(s, r, o)$ tuple form, and KG is defined as $\mathcal{KG} = \{(s, r, o)\} \subset \mathcal{E} \times \mathcal{R} \times \mathcal{E}$, where $s$ is a string representing subject, $r$ is a string representing relation, and $o$ is a string representing object. $\mathcal{E}$ is a set of strings representing entities, and $\mathcal{R}$ is a set of strings representing relationships. An example of a sentence corresponding to the knowledge $(s, r, o)$ such that $s =$ "Donald John Trump", $o =$ "NewYork", $r =$ "born in" is "Donald John Trump was born in NewYork.". An overview of KG is shown in Fig. 1. Here, KG is drawn with entities as nodes and relations as edges. When constructing a KG of LLM, we mask a portion of the above knowledge $(s, r, o)$ and convert it into a prompt. If the LLM can correctly answer the masked prompt,

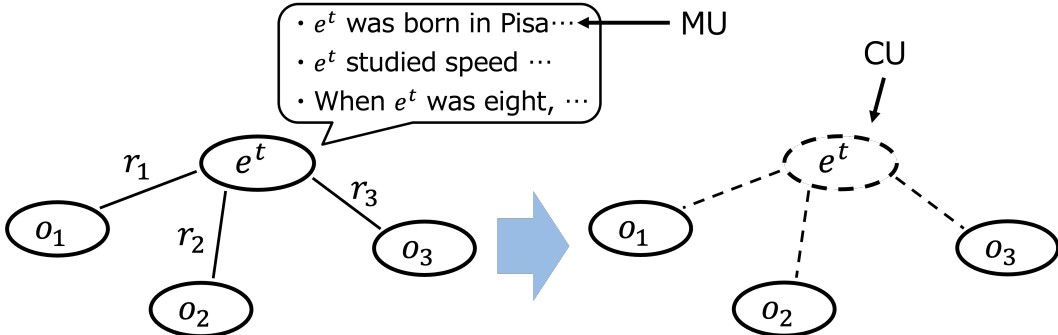

Figure 1: **The overview of CU and differences between CU and MU.** CU requires removing the forgetting target node and the edges of other nodes connected to the target node from the KG of the LLM. On the other hand, MU requires removing the knowledge of some target sentences and does not consider KG.

the LLM is considered to possess the above knowledge [Petroni et al., 2019, Luo et al., 2023]. In this paper, we consider the LLM forgetting based on KG.

### 2.2.2 Machine Unlearning

Machine Unlearning (MU) is a task that aims to forget the arbitrary training data from the AI model [Bourtoule et al., 2021, Ginart et al., 2019, Yao et al., 2023, Neel et al., 2021]. The MU problem setting is shown below. Let $\mathcal{Z}$ be the data point space and $2^{\mathcal{Z}}$ be the power set of $\mathcal{Z}$. Let $D \in 2^{\mathcal{Z}}$ be the training dataset sampled from $\mathcal{Z}$ and $\mathcal{H}$ be the hypothesis set of the AI model. Also, define $D_f \subset D$ as the forgetting target dataset. Let $\mathcal{A}$ be a learning algorithm and $\mathcal{U}$ be the forgetting algorithm. Let $\mathcal{A}$ and $\mathcal{U}$ be stochastic algorithms, respectively, and let $\theta^* \sim \mathcal{A}(D)$ be the trained model. The requirement for MU is that the probability distributions of $\mathcal{A}(D \setminus D_f)$ and $\mathcal{U}(D_f, \theta^*)$ are identical.

As seen from the problem setting above, when applying MU, it is necessary to define a forgetting target dataset $D_f \subset D$. However, there may be cases where the deletion request to LLM is for a specific concept, not a specific data point. In such cases, it is difficult to specify the forgetting target dataset in the training dataset, and the existing MU problem setting may not be able to handle the forgetting of the target concept. Therefore, a different requirement for forgetting is needed to meet the concept deletion requirement for LLM.

### 2.2.3 Knowledge Edit

Knowledge Edit (KE) is a research area that aims to edit the knowledge possessed by LLMs by modifying the model parameters [Meng et al., 2022, Dai et al., 2022]. In these studies, the knowledge possessed by LLMs is defined by KG, and the goal of KE is to edit the objective on the specific knowledge $(s, r, o)$ to $(s, r, o')$. For example, if we edit the knowledge ("Donald John Trump", "born in", "NewYork") to ("Donald John Trump", "born in", "Seattle"), the LLM output will be "Donald John Trump was born in Seattle.". KE differs from MU in that the purpose of KE is to edit the knowledge $(s, r, o)$ on the KG of LLM, not to forget some knowledge.

## 3  Concept Unlearning

CU is a new requirement for MU that seeks to remove forgetting target entities from the KG of the LLM. An overview of CU is shown in Fig. 1. Defining $e^t$ as the string representing the forgotten entity, the knowledge containing the forgotten entity can be written as $(e^t, r, o)$ or $(s, r, e^t)$, where $o \neq e^t$ and $s \neq e^t$. To delete the forgetting target entity on KG of the LLM, it is necessary to delete the node of $e^t$ and the edges of other nodes connected to the forgetting target node. These requirements for KG can be reduced to the following two requirements for CU.

**Requirement 1 (Node Unlearning): Do not output the forgetting target entity.**
The deletion of the target node on the KG of LLM corresponds to not outputting $e^t$ when the LLM is

given a prompt masking the token sequence $e^t$ for the knowledge containing the forgetting target entity $((e^t, r, o)$ or $(s, r, e^t))$ [Petroni et al., 2019]. In this paper, we call this requirement Node Unlearning (NU) and formulate it as follows

$$\forall \boldsymbol{x} \in V^N \left( \boldsymbol{e}^t \not\subset \boldsymbol{x} \rightarrow \boldsymbol{e}^t \not\subset \boldsymbol{f}_\theta(\boldsymbol{x}) \right), \tag{2}$$

where $\boldsymbol{e}^t$ is a tokenized token sequence of the target entity. This formula specifies that the LLM does not output $\boldsymbol{e}^t$ when an input token sequence $\boldsymbol{x}$ does not contain $\boldsymbol{e}^t$. It corresponds to the removal of the target node from the KG of the LLM.

**Requirement 2 (Edge Unlearning): Do not explain the forgetting target entity well.**
The deletion of edges connecting to the forgetting target node on the KG of the LLM means that when the LLM is given a prompt that masks the token sequence other than the forgetting target entity $\boldsymbol{e}^t$, the LLM does not output the correct knowledge containing the forgetting target $((e^t, r, o)$ or $(s, r, e^t))$ [Petroni et al., 2019]. In this paper, we call this requirement Edge Unlearning (EU) and formulate it as follows

$$\forall \boldsymbol{x} \in V^N \left( \boldsymbol{e}^t \subset \boldsymbol{x} \rightarrow [\boldsymbol{x}, \boldsymbol{f}_\theta(\boldsymbol{x})] \notin U^t \right), \tag{3}$$

where $[\boldsymbol{x}, \boldsymbol{f}_\theta(\boldsymbol{x})]$ is a token sequence combining the input and output of the LLM, and let $U^t$ be a set of token sequences representing the correct knowledge containing the forgetting target entity $((\boldsymbol{e}^t, \boldsymbol{r}, \boldsymbol{o}), (\boldsymbol{s}, \boldsymbol{r}, \boldsymbol{e}^t))$. This formula specifies that when a forgetting target entity $\boldsymbol{e}^t$ is input to the LLM, no other entities connected to $\boldsymbol{e}^t$ are output, corresponding to the removal of edges between the target node and the other nodes on the KG of the LLM.

## 4 Proposal

In this chapter, we propose a method to realize CU. The proposed method aims to realize CU by performing GA on the LLM using appropriate token sequences for each CU requirement. The appropriate token sequences are generated by cooperating the forgetting target LLM and another LLM (reference LLM). In this chapter, we first describe the functions and GA used in our method as base functions, and then explain the forgetting process for each CU requirement.

### 4.1 Base functions

First, we define the MASK function and the MAKE_PROMPT function.
$\mathrm{MASK}(\boldsymbol{x}, \boldsymbol{e}^t)$: In the token sequence $\boldsymbol{x}$, replace the substring matching the token sequence $\boldsymbol{e}^t$ representing the forgetting target entity with a [MASK] token and return it.
$\mathrm{MAKE\_PROMPT}(\boldsymbol{x}, \boldsymbol{e}^t)$: Using the MASK function, replace the substring of the token sequence $\boldsymbol{x}$ that matches the forgetting target entity $\boldsymbol{e}^t$ with a [MASK] token, and then create a prompt that asks what the token sequence is supposed to describe and return it. The algorithm of the MAKE_PROMPT function is shown in Algorithm 1.

GA is a learning algorithm that updates the model parameters of an AI model in the direction of climbing with respect to the loss. GA updates the parameters in the opposite direction of Gradient Descent, a common learning algorithm, and can be written as follows.

$$\boldsymbol{\theta} \leftarrow \boldsymbol{\theta} + \lambda \nabla_{\boldsymbol{\theta}} \mathcal{L}(\boldsymbol{x}; \boldsymbol{\theta}), \tag{4}$$

where $\boldsymbol{\theta}$ are the model parameters of the AI model, $\lambda$ is the learning rate, and $\mathcal{L}$ is the loss with respect to the data $\boldsymbol{x}$. When GA is applied to the model, the model parameters are updated to increase the loss with respect to the given data $\boldsymbol{x}$, thus degrading the accuracy of the data $\boldsymbol{x}$. In MU that aims to forget the arbitrary training data, a method based on GA has been proposed Neel et al. [2021], Yao et al. [2023]. In this paper, we aim to realize CU by performing GA several times on the appropriate token sequences for the two CU requirements.

### 4.2 Forgetting algorithm for NU

NU specifies that the LLM will no longer output a token sequence representing the forgetting target entity; by relaxing the definition formula for NU, we obtain the following formula.

$$\forall \boldsymbol{x} \in V^N \left( \boldsymbol{e}^t \not\subset \boldsymbol{x} \rightarrow \sum_{i=1}^{M-|e^t|} \prod_{j=1}^{|e^t|} p_\theta(e_j^t | \boldsymbol{x}, \boldsymbol{y}_{\mathrm{out},i}, \boldsymbol{e}_j^t) \ll 1 \right), \tag{5}$$

---

**Algorithm 1** MAKE_PROMPT function

---

**Require:** $\boldsymbol{x}, \boldsymbol{e}^t$
1: **if** $\boldsymbol{e}^t \subset \boldsymbol{x}$ **then**
2:     $\boldsymbol{x}_{\text{mask}} = \text{MASK}(\boldsymbol{x}, \boldsymbol{e}^t)$
3:     $\boldsymbol{x}_{\text{prompt}} = $ "Tell me [MASK] in the following. $\{\boldsymbol{x}_{\text{mask}}\}$"
4: **else**
5:     $\boldsymbol{x}_{\text{prompt}} = $ "Tell me what the following says. $\{\boldsymbol{x}_{\text{mask}}\}$"
6: **end if**
7: **return** $\boldsymbol{x}_{\text{prompt}}$

---

---

**Algorithm 2** Forgetting algorithm for NU

---

**Require:** $\boldsymbol{e}^t$, $\boldsymbol{x}_{\text{pro}}=$"Tell me about $\{\boldsymbol{e}^t\}$.", $\theta$, $\theta_{\text{ref}}, \lambda$
1: $\boldsymbol{y}_{\text{ref}} = \boldsymbol{f}_{\theta_{\text{ref}}}(\boldsymbol{x}_{\text{pro}})$
2: $\boldsymbol{x}_{\text{ref}} = \text{MAKE\_PROMPT}(\boldsymbol{y}_{\text{ref}}, \boldsymbol{e}^t)$
3: $\boldsymbol{y} = \boldsymbol{f}_\theta(\boldsymbol{x}_{\text{ref}})$
4: **if** $\boldsymbol{e}^t \subset \boldsymbol{y}$ **then**
5:     $\theta \leftarrow \theta + \lambda \nabla_\theta \mathcal{L}_1(\boldsymbol{y}_{\text{ref}}; \theta)$
6: **end if**

---

---

**Algorithm 3** Forgetting algorithm for EU

---

**Require:** $\boldsymbol{e}^t$, $\boldsymbol{x}_{\text{pro}}=$"Tell me about $\{\boldsymbol{e}^t\}$.", $\theta$, $\theta_{\text{ref}}, \lambda$
1: $\boldsymbol{y} = \boldsymbol{f}_\theta(\boldsymbol{x}_{\text{pro}})$
2: $\boldsymbol{x}_{\text{ref}} = \text{MASK}(\boldsymbol{y}, \boldsymbol{e}^t)$
3: $\boldsymbol{y}_{\text{ref}} = \boldsymbol{f}_{\theta_{\text{ref}}}(\boldsymbol{x}_{\text{ref}})$
4: **if** $\boldsymbol{e}^t \subset \boldsymbol{y}_{\text{ref}}$ **then**
5:     $\theta \leftarrow \theta + \lambda \nabla_\theta \mathcal{L}_2(\boldsymbol{y}; \theta)$
6: **end if**

---

where $\boldsymbol{y}_{\text{out},i} = [y_1, \cdots, y_i]$ represents the first to $i$-th of the LLM output token sequence, and $\boldsymbol{e}_j^t = [e_1, \ldots, e_j^t]$ represents the first to $j$-th of the token sequence representing the forgetting target entity. Eq 5 expresses that the probability of outputting a token sequence $\boldsymbol{e}^t$ representing the forgetting target entity at any position in the output token sequence is as close to 0 as possible. If Eq. 5 is satisfied, the LLM does not output the forgetting target entity $\boldsymbol{e}^t$, and the NU requirement is satisfied. We define $\mathcal{L}_1$ as the loss function as follows.

$$\mathcal{L}_1(\boldsymbol{x}; \boldsymbol{\theta}) = -\log \sum_{i=1}^{M-|e^t|} \prod_{j=1}^{|e^t|} p_{\boldsymbol{\theta}}(e_j^t | \boldsymbol{x}, \boldsymbol{y}_{\text{out},i}, \boldsymbol{e}_j^t). \tag{6}$$

Next, we consider the token sequences for the NU forgetting algorithm. In order to efficiently realize NU by GA, it is desirable to obtain token sequences such that the expression 5 takes small values, i.e., the input token sequence that has a high probability of outputting $\boldsymbol{e}^t$ for the target LLM. To obtain such a token sequence $\boldsymbol{x}$, we use a reference LLM $\theta_{\text{ref}}$. In our method, the reference LLM is used to obtain a description of the forgetting target entity. Therefore, the reference LLM is required to know the forgetting target entity.

When obtaining the token sequence for the NU forgetting algorithm, the prompt $\boldsymbol{x}_{\text{pro}} =$"Tell me about $\{\boldsymbol{e}^t\}$." is given to the reference LLM $\theta_{\text{ref}}$ to obtain $\boldsymbol{y}_{\text{ref}}$, which is a description of the forgetting target entity. Then, by applying the MAKE_PROMPT function to the obtained explanatory text $\boldsymbol{y}_{\text{ref}}$, a prompt $\boldsymbol{x}_{\text{ref}}$ is generated to ask what $\boldsymbol{y}_{\text{ref}}$ explains. The prompt $\boldsymbol{x}_{\text{ref}}$ is then input to the target LLM, and the output is checked to see if $\boldsymbol{e}^t$ is included. If $\boldsymbol{e}^t$ is included in the output of the target LLM, $\boldsymbol{y}_{\text{ref}}$ is used as the token sequence for the NU forgetting algorithm. This algorithm can perform GA on the token sequence $\boldsymbol{y}_{\text{ref}}$ that has a high probability of outputting $\boldsymbol{e}^t$ for the target LLM to and can realize NU efficiently. NU forgetting algorithm is shown in Algorithm 2.

### 4.3 Forgetting algorithm for EU

EU specifies that when the LLM is given a token sequence representing the forgetting target entity, it should not output a token sequence representing other entities associated with the forgetting target entity. By relaxing Eq 3, we obtain the following formula.

$$\forall \boldsymbol{x} \in V^N, \forall \boldsymbol{y} \in U^t, \left( p_{\boldsymbol{\theta}}(\boldsymbol{y}|\boldsymbol{x}) = \prod_{i=1}^M p_{\boldsymbol{\theta}}(y_i | \boldsymbol{x}, y_1, \ldots, y_{i-1}) \ll 1 \right). \tag{7}$$

> "subject"    : Galileo Galilei,
>
> "sentences": ["[MASK] was born in Pisa …",
>
>                         "[MASK] studied speed and velocity, …",
>
>                         "When [MASK] was eight, his family moved …", ]

Figure 2: **Wiki-Person dataset.**    A dataset consisting of entities to be forgotten (subjects) and sentences about the entities to be forgotten (sentences). The character string representing the target to be forgotten in the sentences is replaced by [MASK].

Equation 7 expresses that for any input, the LLM does not output a token sequence representing correct knowledge about the forgetting target entity. Therefore, if Eq 7 is satisfied, the EU requirement that no token sequence representing other entities related to the forgetting target entity is satisfied. The above equation defines the loss $\mathcal{L}_2$ in the following form.

$$
\begin{aligned}
\mathcal{L}_2(\boldsymbol{x}; \boldsymbol{\theta}) \quad &= -\log \prod_{i=1}^{M} p_{\boldsymbol{\theta}}(y_i|\boldsymbol{x}, y_1, \cdots, y_{i-1}) \\
&= -\sum_{i=1}^{M} \log p_{\boldsymbol{\theta}}(y_i|\boldsymbol{x}, y_1, \cdots, y_{i-1}).
\end{aligned}
\tag{8}
$$

Next, we consider the token sequences for the EU forgetting algorithm. To efficiently realize EU by GA, it is desirable to use token sequences included in $U^t$, i.e., sentences that explain the correct knowledge including the forgetting target entity. To select such explanatory sentences, we use the reference LLM $\theta_{\mathrm{ref}}$.

In the process for EU, the target LLM is first given $\boldsymbol{x}_{\mathrm{pro}} =$"Tell me about $\{\boldsymbol{e}^t\}$." and obtain the output $\boldsymbol{y}$. Then, the output $\boldsymbol{y}$ is processed in the MAKE_PROMPT function to generate a prompt $\boldsymbol{y}_{\mathrm{ref}}$ that asks what $\boldsymbol{y}$ explains about. $\boldsymbol{y}_{\mathrm{ref}}$ is then input to the reference LLM, which checks whether $\boldsymbol{y}$ is in $U_t$ or not. If $\boldsymbol{e}^t$ is included in the output of the reference LLM, we judge that $\boldsymbol{y}$ is in $U^t$ and is used as the token sequence for the EU forgetting algorithm. This algorithm allows us to learn the target LLM to avoid the output of the token sequence contained in $U^t$, effectively achieving EU. The forgetting algorithm for EU is shown in Algorithm 3.

## 5    Experimental Setup

We describe the dataset used in the experiment. We made a dataset of forgetting target entities and their descriptions from Wikipedia. The forgetting target entity in this experiment is a person's name. The Wikipedia article about the forgetting target person's name is divided into paragraphs, and the paragraphs that contain the person's name are collected and used as data. In addition, the strings of the forgetting target person's name in the descriptions are replaced with [MASK]. We show a part of the dataset in Fig. 2. The dataset contains the names of 20 target persons. Hereafter, this dataset is referred to as the Wiki-Person dataset. We also use 8 datasets for evaluating general knowledge of the LLM (HellaSwag, Lambada, Winogrande, COPA, ARC-Easy, ARC-Challenge, MathQA, and PubmedQA). HellaSwag and Lambada are datasets of linguistic reasoning ability, winogrande and COPA are datasets of commonsense-based reasoning ability, and ARC-Easy, ARC-Challenge, MathQA, and PubmedQA are datasets of scientific reasoning ability. Using these datasets, we evaluate if our method damages the general performance of the target LLM [Jang et al., 2022].

### 5.1    Implementation

The LLMs used in the evaluation experiments are the Mistral-7B Instruction model [Jiang et al., 2023] and the Llama3-8B model [Touvron et al., 2023]. In addition, the Mistral-7B Instruction model is used as the reference LLM in the forgetting process. The learning rate of GA in the forgetting process is set to $10^{-7}$, and Adam is used as the learning algorithm. In the evaluation experiment, the forgetting algorithm corresponding to each of the CU requirements is performed 10 times alternately. The subject of the Wiki-Person dataset is used as the forgetting target person's name, and the forgetting performance is evaluated after applying the forgetting process to each subject. In the evaluation experiments, we do not evaluate the person names of which the LLM already satisfies CU requirements before the forgetting process (i.e., the person that the LLM does not know from the beginning).

Table 1: Forgetting evaluation for our method.

(a) Mistral-7B Instruction model.

|  | Viorate for NU ↓ | Viorate for EU ↓ |
|---|---|---|
| Pre-forget | $0.25 \pm 0.09$ | $0.70 \pm 0.29$ |
| Post-forget | $\mathbf{0.1 \pm 0.15}$ | $\mathbf{0.00 \pm 0.00}$ |

(b) Llama3-8B model.

|  | Viorate for NU ↓ | Viorate for EU ↓ |
|---|---|---|
| Pre-forget | $0.21 \pm 0.05$ | $0.74 \pm 0.32$ |
| Post-forget | $\mathbf{0.12 \pm 0.07}$ | $\mathbf{0.31 \pm 0.32}$ |

## 5.2 Evaluation Metrics

**Evaluation for each CU requirement**

To evaluate the NU requirement, we input the LLM the task of answering [MASK] in the description of the Wiki-Person dataset. Then, we evaluate whether the LLM outputs the forgetting target entity. The prompt given to the LLM is $\mathrm{MAKE\_PROMPT}(\boldsymbol{x}, \boldsymbol{e}^t)$, where $\boldsymbol{x}$ is a description in the Wiki-Person dataset and $\boldsymbol{e}^t$ is a token sequence representing the forgetting target entity.

When evaluating the EU requirement, the LLM is given a prompt ("Tell me about $\{\boldsymbol{e}^t\}$.") to output a description of the forgetting target entity. Then, the MAKE_PROMPT function is applied to the explanatory text $\boldsymbol{x}$ output from the LLM and obtain the prompt $\boldsymbol{y} = \mathrm{MAKE\_PROMPT}(\boldsymbol{x}, \boldsymbol{e}^t)$, which asks what the explanatory text is about. The obtained prompt $\boldsymbol{y}$ is input to GPT-4o to answer what $\boldsymbol{y}$ explains and check if the answer of GPT-4o contains the forgetting target entity. In this process, we evaluate whether the LLM outputs a token sequence about the correct knowledge $((\boldsymbol{e}^t, \boldsymbol{r}, \boldsymbol{o}),$ $(\boldsymbol{s}, \boldsymbol{r}, \boldsymbol{e}^t))$ involving the forgetting target entity in the manner of LLM-as-a-judge Zheng et al. [2024].

**Evaluation of LLMs for general performance.**

To evaluate the general performance of LLMs, we test them with the 8 inference tasks (HellaSwag, Lambada, Winogrande, COPA, ARC-Easy, ARC-Challenge, MathQA, PubmedQA). This evaluation is performed on LLMs with and without our forgetting process to see how our method affects the performance of LLMs in general.

## 6 Experimental Results

The performance of the CU requirements is shown in Tables 1a and 1b. For each requirement, the rate of outputs that violate the requirement is measured for each forgetting target entity, and the table shows the mean and standard derivation in the entity direction. Our forgetting process for both the Mistral-7B-Instruction model and the Llama3-8B model significantly reduces the rate of outputs that violate the requirement. Tables 2a and 2b show the output obtained by inputting the prompt "Tell me about Galileo Galilei. If you don't know, say I don't know." to the Mistral-8B-Instruction model before and after our forgetting process about Galileo Galilei. Table 2a confirms that the LLM outputs the correct description of Galileo Galilei before forgetting that he was an Italian physicist, mathematician, astronomer, and philosopher and that he is regarded as the father of observational astronomy. On the other hand, from Table 2b, we can confirm that the LLM answers "I don't know." and describes a person other than Galileo Galilei, and does not say about Galileo Galilei.

These results confirm that our method can make the LLM that satisfies each CU requirement to some extent. However, from Tables 1a and 1b, it can also be confirmed that both the Mistral-7B-Instruction model and the Llama3-8B model are not able to reduce the rate of outputs that violate the CU requirements to zero. Tables 1a and 1b, also show that the standard deviation is large. Looking at the individual forgetting results, we can see that the LLM can satisfy the CU requirements for some of the forgetting target entities, while it cannot satisfy them for the other forgetting target entities. It is a subject for future work to reduce the standard deviation in forgetting performance due to differences in forgetting target entities by developing a method. In this study, the Mistral-8B-Instruction model was used as the reference LLM, but the forgetting performance of our method is expected to vary depending on the performance of the reference LLM. The search for a reference LLM suitable for the forgetting process is also a future work. Also, Table 2b shows that our method can eliminate the description of Galileo Galilei, but it describes a different person, which is a hallucination. This result indicates that our GA-based method may generate LLMs that cause hallucinations about the forgetting target entity, and there is still room for debate as to what kind of LLMs should be aimed for the forgetting model.

Table 2: LLM output when asked about Galileo Galileo.

(a) Pre-forgotten LLM.

| Output |
| --- |
| Galileo Galilei (1564-1642) was an Italian physicist, mathematician, astronomer, and philosopher who played a major role in the Scientific Revolution. |
| He is considered to be the father of observational astronomy and is best known for his improvements to the telescope and his astronomical discoveries. |

(b) Post-forgotten LLM.

| Output |
| --- |
| I don't know. Galen Leo Galen or Galenus, known as Galen of Pergamon, was a Greek physician, philosopher, and scientist, considered one of the most influential figures in the history of medicine. |
| He was born in Pergamon, a city in the Roman province of Asia Minor, in 12 or 13 February AD 150 and died in February AD 216. |

Table 3: Performance Evaluation for Mistral-7B-Instruction model.

|        | HellaSwag ↑ | Lambada ↑ | Winogrande ↑ | COPA ↑ | ARC-Easy ↑ | ARC-Challenge ↑ | MathQA ↑ | PubmedQA ↑ |
| --- | --- | --- | --- | --- | --- | --- | --- | --- |
| Before | 0.56 | 0.70 | 0.66 | 0.88 | 0.88 | 0.77 | 0.14 | 0.38 |
| After  | 0.57 | 0.67 | 0.66 | 0.87 | 0.88 | 0.77 | 0.14 | 0.38 |

Tables 3 and 4 show the evaluation results of the general performance of LLM before and after forgetting. These tables show that there is no significant degradation in general performance for our forgetting process. Our method generates appropriate token sequences for forgetting and applies GA to them. Therefore, we believe that the damage to the remained concepts can be kept small, and as a result, the degradation of the general performance of the remained is minimized.

## 7 Conclusion

In this paper, we proposed CU as a new requirement for MUs to respond to deletion requests for LLMs. In addition, we proposed an approach to realize CU by performing GA on appropriate token sequences. For the evaluation, we created a person name dataset (Wiki-person dataset) from Wikipedia and confirmed that our method can suppress the output of the forgetting target entities to some extent. However, it was also confirmed that our method promotes hallucination, and there is still room for debate as to what kind of LLM should be aimed at forgetting. We also confirmed that there are cases where CU cannot be fully realized depending on the forgetting target, and we will discuss the improvement of these cases in the future.

Table 4: Performance Evaluation for Llama3-8B model.

|        | HellaSwag ↑ | Lambada ↑ | Winogrande ↑ | COPA ↑ | ARC-Easy ↑ | ARC-Challenge ↑ | MathQA ↑ | PubmedQA ↑ |
| --- | --- | --- | --- | --- | --- | --- | --- | --- |
| Before | 0.38 | 0.70 | 0.52 | 0.57 | 0.43 | 0.38 | 0.17 | 0.02 |
| After  | 0.38 | 0.70 | 0.52 | 0.56 | 0.46 | 0.41 | 0.17 | 0.03 |

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
