# OpenReview forum: "Concept Unlearning for Large Language Models"
_NeurIPS.cc/2024/Workshop/SafeGenAi — SafeGenAi Poster_

### Official Review · Reviewer_4ASS · 2024-10-09
**This paper proposes a novel approach, Concept Unlearning, to achieve machine learning with a knowledge graph and removing concepts. The method's effectiveness is validated with solid experiments. I believe this paper should be accepted.**

**Rating:** 7
**Confidence:** 3

**Review:**

The paper introduces a novel approach, Concept Unlearning (CU) to realize machine unlearning through removing concepts, like a person's name, from large language models (LLMs) using knowledge graphs. This method shows its superiority over traditional unlearning method, which can only delete specific training data points, but may improves upon traditional unlearning, which only targets specific data points and suffers from hallucination.  The experiment results shows promising results in handling data privacy concerns and maintaining general knowledge. Maybe in the future, the authors can further work on problems arise from the experiment results, like hallucination, incomplete forgetting, and dependence on reference LLM to further improve the current work.

---

### Official Review · Reviewer_m2Fa · 2024-10-09
**Review for Submission 44**

**Rating:** 7
**Confidence:** 3

**Review:**

This work focus on the concept unlearning (CU) for LLMs, and propose a novel method for CU. The experiment results show that it can forget the target concepts well while remaining performance on normal benchmarks. The paper is clearly written, and point out the limitation of previous works on machine unlearning. Overall, I think this is a good paper and it is highly relevant to the workshop.

This paper also honestly tell the limitation of the paper, which I appreciate. I agree that concept unlearning will increase hallucination, but hallucination is the feature of current LLMs. Therefore, I think this limitation is out-of-scope for this paper.
My questions and weakness:
* For Algorithm 1, why only two types of $**x**_{prompt}$ is sufficient for unlearning.
* how to compute the loss, $p_{\theta}$ for $L_1$ and $L_2$ is unclear.

---

### Official Review · Reviewer_GGUN · 2024-10-09
**There are some ideas but I don't see the practicality**

**Rating:** 5
**Confidence:** 4

**Review:**

### Summary
The paper proposes a method called Concept Unlearning (CU) for Large Language Models (LLMs) to forget specific concepts using knowledge graphs and gradient ascent. While the method is novel in its attempt to forget concepts rather than individual data points, it falls short in several key areas that impact its effectiveness and robustness.

### Strengths
1. **Innovative Method:** Introducing Concept Unlearning through the use of knowledge graphs is a promising direction that aims to address privacy concerns in a more human-understandable way.
2. **Algorithm Design:** The use of gradient ascent (GA) to degrade the model's knowledge of specific concepts is clearly defined, with detailed explanations for Node Unlearning (NU) and Edge Unlearning (EU).
3. **Comprehensive Evaluation:** The authors evaluated their method on various datasets, demonstrating some effectiveness in concept forgetting while preserving general knowledge.

### Weaknesses
1. **Lack of Robustness:** From my perspective, the method's performance is inconsistent, showing significant variance depending on the target concept.
2. **Hallucination Issues:** The problem of hallucinations introduced by this method is not adequately addressed.
3. **Dependency on Reference LLM:** The effectiveness of the method relies heavily on a reference LLM, which may limit its generalizability depending on the quality of the reference model.
4. **Incomplete Unlearning:** The method does not fully satisfy Concept Unlearning requirements for all targets, and the standard deviation in results suggests incomplete and inconsistent unlearning, which also complicates the implementation of the method.
5. **Simplistic Methodology:** The method is primarily based on masking the target entity and re-querying it with a new LLM, which lacks depth and sophistication for a concept-unlearning technique.
6. **Unexplored Impact on Remaining Entities:** The method does not analyze the effect of unlearning one entity on other entities or concepts within the model.
7. **Limited Evaluation Scope:** The evaluation focuses exclusively on entities, without extending to other data types or broader concepts, limiting its conclusions on general applicability.

### Recommendation
The method proposed in this paper is innovative but not yet mature enough for practical use due to its lack of robustness, issues with hallucinations, and incomplete unlearning. Given these significant weaknesses and the reliance on a relatively straightforward masking technique, I recommend rejecting the paper with a score of 5. Addressing these concerns and broadening the evaluation could strengthen the method for future submissions.

In addition, the layout of the table4 at the end of page 8 should be modified.